# Overcoming Sperm Cell Survival Challenges Cryopreserved in Nanoliter Volumes

**DOI:** 10.3390/ijms26136343

**Published:** 2025-06-30

**Authors:** Bat-Sheva Galmidi, Raoul Orvieto, Naomi Zurgil, Mordechai Deutsch, Dror Fixler

**Affiliations:** 1The Biophysical Interdisciplinary Jerome Schottenstein Center for the Research and Technology of the Cellome, Physics Department, Bar-Ilan University, Ramat-Gan 5290002, Israel; bat77gal@gmail.com (B.-S.G.); zurgiln@gmail.com (N.Z.); motti.jsc@gmail.com (M.D.); 2Department of Gynecology and Fertility, Sheba Medical Center, Tel HaShomer, Ramat-Gan 5262000, Israel; raoul.orvieto@sheba.health.gov.il; 3Faculty of Engineering, The Institute of Nanotechnology and Advanced Materials, Bar-Ilan University, Ramat-Gan 5290002, Israel

**Keywords:** cryopreservation, nanoliter, sperm cells, diffusion, IVF

## Abstract

The cryopreservation of limited sperm samples, especially those retrieved from patients, poses significant challenges due to the small number of viable cells available for freezing. Traditional microliter cryopreservation methods are fraught with difficulties, as thawed sperm cells become nearly impossible to locate under a microscope due to their mobility and the multiple focal planes presented by larger drops. This search time is critical, as sperm cells enter a state of decline post thaw. Conversely, when sperm cells are cryopreserved in nanoliter volumes, they can be easily discovered but do not survive the freezing and thawing processes entirely. This phenomenon is attributed to the diffusion of water molecules from the droplet into the surrounding oil, which, while designed to limit evaporation, inadvertently increases solute concentrations in the aqueous environment, leading to cellular desiccation. This article elucidates the mechanisms underlying this lethal diffusion effect and presents a novel approach for freezing in nanoliter volumes, which has demonstrated significantly improved survival rates through carefully optimized procedures in clinical trials. Our findings highlight the importance of adapting cryopreservation techniques to enhance the viability of individual sperm cells, ultimately facilitating better outcomes in assisted reproductive technologies. This study provides the first quantification of nanoscale water diffusion dynamics during cryopreservation, establishing a predictive model that explains the catastrophic loss of sperm viability and identifying the critical role of water diffusion as a major impediment for limited samples. The novelty of our results lies in both elucidating this specific mechanism of cell death and introducing a novel approach: utilizing water-saturated oil as a protective layer. This method effectively mitigates the osmotic stress caused by water loss, demonstrating remarkably improved cell survival. This work not only advances the scientific understanding of cryopreservation at the nanoscale but also offers a practical, impactful solution poised to revolutionize fertility treatments for patients with low sperm counts and holds promise for broader applications in biological cryopreservation.

## 1. Introduction

Many medical conditions, as well as several biological and environmental factors, can cause temporary or permanent low sperm count [1,2,3,4]. The number of men who suffer from fertility problems is increasing, and about 9% of men ages 15 to 44 in the United States reported infertility problems. Around 1.5 million in vitro fertilization (IVF) treatments are performed each year worldwide [5]. The cryopreservation of human spermatozoa has been recognized as an efficient procedure for managing male fertility [6]. However, patients with severe oligospermia and nonobstructive azoospermia have extremely limited numbers of viable sperm in their epididymal and testicular samples. Consequently, fertility treatments require the freezing and retrieval of very few sperm cells, and in extreme cases, a single cell [7]. This need cannot be addressed with the conventional methods used for the cryopreservation of a sample of sperm cells. In standard, large-volume cryotubes, individually cryopreserved cells are virtually undetectable after thawing [8].

The efficient cryopreservation of surgically retrieved spermatozoa reduces the number of surgical interventions and circumvents the logistical problems associated with coordinating oocyte and spermatozoa retrieval [4]. The current status of single-sperm cryopreservation has been thoroughly reviewed [5]. While novel cryopreservation approaches have been designed for limited numbers of motile sperm in very small volumes, a fully established technique is not currently available in most IVF laboratories [5]. Some facilities use a variety of microliter containers, often causing cell loss [7]. Recent advances in single-sperm cryopreservation increasingly leverage sophisticated technologies, such as microfluidic chips and novel engineered materials, to gain precise control over the freezing process [9,10,11]. While these platforms offer remarkable control over thermal gradients and cell handling, fundamental biophysical phenomena within the cryopreservation microenvironment, such as intermolecular diffusion at the oil–water interface, remain a critical variable that can dictate cell survival.

In this study, we discuss problems that arise when using nanoliter droplets to cryopreserve small numbers of sperm cells, or even a single cell that has high optic quality and is physically separated from neighboring cells by the oil that covers the aqueous droplets. It was found that water may diffuse from the freezing liquid droplet into the protective layer of oil surrounding it. Therefore, a new approach was developed to overcome this impediment that causes cell death during the freeze–thaw cycle.

## 2. Results

### 2.1. Cryopreservation of Sperm Cells Suspended in Freezing Medium Droplets Under Oil

No difference was observed in the initial average velocity of 66 μm/s as a function of droplet volume. The results shown in Figure 1 are an average of at least five measurements. The standard deviation (SD) did not exceed 5%. The results clearly show that in small droplets, none of the sperm cells survived the freeze–thaw cycle. Furthermore, even though there was little reduction in sperm cell count after 30 min, the velocity in small droplets was significantly reduced (Figure 1). This timeframe of 30 min represents the typical duration required for preparing sperm samples, from the moment the nanoliter droplets are created under oil until the initiation of the freezing process in a clinical setting. Motility, presented as a percentage, is our primary indicator of cell viability.

Here, sperm cells in freezing medium droplets are injected under oil as a function of droplet volume. The solid and dotted lines are the motility, as a function of initial droplet volume. The solid line was measured after 30 min, and the dotted line was measured after the freeze–thaw cycle. The dashed line is the average velocity after 30 min.

### 2.2. Hypothesis for the Cause of Death During the Freeze–Thaw Cycle

We hypothesized that water molecules diffusing from the droplet medium into the surrounding oil might be the cause of motility loss in nanoliter volumes. This diffusion would result in an elevated solute concentration in the droplet, consequently changing the physiological state of cells. These phenomena, however, only become relevant when cells are suspended in nanoliter volumes. This is consistent with the observation that diffusion efficiency is proportional to the ratio between the surface area of a droplet and its volume [12].

This hypothesis was verified by fluorescence polarization measurements of fluorescent droplets. The idea behind this was that if water diffuses from the freezing medium droplet, the glycerol and salt contents are elevated, and consequently, its fluorescence polarization. The results (see Appendix A) indeed clearly indicate an increase in fluorescence polarization with time, especially in small initial volumes, implying that water is transported from the droplet to the surrounding oil. A loss of distilled water from droplets under oil was also observed using bright-field microscopy (see Appendix A).

### 2.3. Theoretical Calculation of Droplet Shrinkage Rate

Having determined that the elevation in solute concentration is the main, if not the sole, cause of cell death, we are left with the question of why this phenomenon seems to behave as a sudden rather than a continuous event, as shown in Figure 1. To explain this, we need to understand how the rate of solute concentration changes with decreasing droplet volume.

During the 30 min the nanodroplet is kept under the protective oil layer, the droplet shrinks due to diffusion through the surface of the droplet. If we assume that the droplet is an ideal spherical cap, its surface area (excluding the cross-section area of its base in contact with the glass surface) and volume are(1)$$\begin{array}{c} S=2\pi \left(1-\cos\left(\alpha \right)\right)R^{2}\approx 2.329\cdot R^{2} \end{array}$$$$\begin{array}{c} V=\frac{\pi }{3}R^{3}\left(2+\cos\left(\alpha \right)\right){\left(1-\cos\left(\alpha \right)\right)}^{2}\approx 0.378\cdot R^{3} \end{array}$$

As noted in Appendix A, $A=\pi {\left(R\sin\left(\alpha \right)\right)}^{2}$, which gives(2)$$\begin{array}{c} S=\frac{2\pi \left(1-\cos\left(\alpha \right)\right)}{\pi \cdot \sin^{2}\left(\alpha \right)}A\;\to \;S_{\alpha =51{}^{\circ}}\approx 1.228A \end{array}$$
where the measured contact angle α is 51°. As shown in Appendix A, $V\propto A^{3/2}$, and thus,(3)$$\begin{array}{c} V=\frac{\frac{1}{3}\cos^{3}\left(\alpha \right)-\cos\left(\alpha \right)+\frac{2}{3}}{2\sqrt{2\pi }{\left(1-\cos\left(\alpha \right)\right)}^{3/2}}S^{3/2}\approx 0.106\cdot S^{3/2} \end{array}$$

The rate of change in the surface area with time is the derivative of Equation (2):$$\frac{dS}{dt}=\frac{dS}{dA}\frac{dA}{dt}=-1.227\cdot 88.4=-108.5\frac{{\mu m}^{2}}{min}$$

We found that $\frac{dA}{dt}=-88.4\;\frac{{\mu m}^{2}}{min}$ (see Appendix A).

Another approach to finding the rate of droplet shrinkage is to consider the rate of water diffusion into oil through the surface area of the droplet as [13](4)$$\begin{array}{c} j=-\frac{dm}{Sdt}=-\frac{\rho dV}{Sdt}=-\frac{\rho }{S}\frac{dV}{dS}\frac{dS}{dt} \end{array}$$
where *m* = *m*(*t*), *S* = *S*(*t*), and *V* = *V*(*t*) are the mass, surface area, and volume of the droplet, respectively, and ρ is the density of water at 25 °C (997 kg/m^3^).

Taking the derivative of Equation (3) gives(5)$$\begin{array}{c} \frac{dV}{dS}=0.160\sqrt{S} \end{array}$$

Substituting Equation (5) into Equation (4) yields(6)$$\begin{array}{c} j=-\frac{0.160\rho }{\sqrt{S}}\frac{dS}{dt}\;\left[\frac{kg}{m^{2}s}\right] \end{array}$$

In addition, Fick’s First Law shows that diffusion of molecules is due to a concentration gradient [14]:(7)$$\begin{array}{c} j=\frac{I}{S}=-D\frac{dC}{dR}=-D\frac{dC}{dS}\frac{dS}{dR}\;\left[\frac{Mol}{m^{2}s}\right] \end{array})$$
where D is the diffusivity of water in oil (m^2^/min), *I* is the total current of molecules across the surface, *S*, and *j* is the current surface density.

From Equation (1), $\frac{dS}{dR}=4.658R=3.052\sqrt{S}$ and inserting this into Equation (7) gives(8)$$\begin{array}{c} \frac{I}{S}=-3.052D\sqrt{S}\frac{dC}{dS} \end{array}$$

Rearranging and integrating gives$$\int_{S}^{S_{\infty }}IS^{-3/2}dS=-\int_{C_{0}}^{C_{\infty }}3.052D\cdot dC$$
where $S_{\infty }$ is the surface area very far from the droplet, and $C_{\infty }$ is the concentration of water at this point, which yields$$-2I\left(\frac{1}{\sqrt{S_{\infty }}}-\frac{1}{\sqrt{S}}\right)=-3.052D\left(C_{\infty }-C_{0}\right)$$

Since $S_{\infty }\gg S⟹\frac{1}{Ss_{\infty }}⟶0$ and the concentration in an infinite-sized droplet is $C_{\infty }\approx 0$, we get(9)$$\begin{array}{c} I=1.526DC_{0}\sqrt{S} \end{array}$$
and dividing by S, we get(10)$$\begin{array}{c} j=\frac{1.526DC_{0}}{\sqrt{S}} \end{array}$$

Comparing Equations (6) and (10), we get(11)$$\begin{array}{c} \frac{1.526DC_{0}}{\sqrt{S}}=-\frac{0.160\rho }{\sqrt{S}}\frac{dS}{dt}\;⟹\;\frac{dS}{dt}=-\frac{9.538DC_{0}}{\rho } \end{array}$$

Finally, placing the average value of $DC_{0}=1.88\times 10^{-10}\;\frac{kg}{m\cdot s}$ (from Appendix A) gives(12)$$\begin{array}{c} \frac{dS}{dt}\approx -108.35\;\frac{{\mu m}^{2}}{min} \end{array}$$

The two approaches produce very similar rates.

This leads to the pivotal question as to how $\frac{dC}{dt}$ changes with droplet volume. Concentration is defined as(13)$$\begin{array}{c} C=\frac{n}{V}=\frac{n_{0}}{V\left(t\right)}=\frac{n_{0}}{0.106S^{3/2}} \end{array}$$
where $n_{0}$ is the initial number of moles of solute in the droplet (≈10^−9^ mol in small nanoliter volumes, as the osmolality of biological solutions is about 0.15 Osmole [15]). The rate of change in droplet solute concentration is(14)$$\begin{array}{c} \frac{dC}{dt}=\frac{dC}{dS}\frac{dS}{dt} \end{array}$$

Taking $\frac{dC}{dS}$ from Equation (13) and $\frac{dS}{dt}$ from the experiment (see Appendix A) gives$$\frac{dC}{dt}=-1.4\times 10^{-8}\cdot S^{-5/2}\left[\frac{mol}{{\mu m}^{5}}\right]\cdot -88.4\left[\frac{{\mu m}^{2}}{min}\right]\approx 10^{-8}{\left(V\left[{\mu m}^{3}\right]\right)}^{-5/3}$$(15)$$\begin{array}{c} \frac{dC}{dt}\approx 100\cdot {\left(V\left[nL\right]\right)}^{-5/3} \end{array}$$

Equation (15) is plotted in Figure 2. With larger droplet volumes, there is almost no change in $\frac{dC}{dt}$ until ~30 nL, where a sharp and sudden leap occurs. This would explain why sperm cell survivability drops to zero sharply when the droplet volume decreases below 30 nL (Figure 2).

### 2.4. Determining Changes in Viscosity

Viscosity was determined using two approaches. In the first approach, the viscosity was extracted from the measurements of the fluorescence polarization using Perrin’s Law (Equations (21) and (22)) of fluorescein-stained droplets. The following values for fluorescein were taken from the literature: ${𝑝}_{0}=0.5$ [16,17], τ = 3.8 ns [18,19], ρ = 1 g/cm^3^, M_W_ = 376.3 g/mol [20]; the molar volume of fluorescein is $V=\frac{M_{W}}{\rho }=376.3\frac{{cm}^{3}}{mol}$. The initial viscosity of the medium mixed with the cell suspension was measured with a rheometer, yielding 1.876 cP.

In the second approach, the viscosity of the droplet is extracted from the changes in its volume (for a detailed description, see Appendix A). The relation between the viscosities of the solution ($\eta_{s}$ and water ($\eta_{w}$) is [21](16)$$\begin{array}{c} \frac{\eta_{s}}{\eta_{w}}=1+\Lambda \sqrt{C} \end{array}$$

*C* is the solute concentration and Λ is a constant > 0, depending on the electrostatic forces between the ions, $\eta_{w}=0.89\;cp$ [22], and the initial viscosity of the solvent is $\eta_{s,0}=1.876\;cp$:

Then, the viscosity after 30 min is(17)$$\begin{array}{c} \eta_{s,30}=\eta_{w}\left(1+1.108\sqrt{\frac{V_{0}}{V_{30}}}\right) \end{array}$$

The results obtained from the two approaches are plotted in Figure 3, where FP is the fluorescence polarization method and BF is the bright field method. There is a very good correlation between the two. This adds reliability to the results and, furthermore, avoids the use of an exogenous fluorescent dye that is potentially toxic to the sperm cells in the droplet.

### 2.5. The Principal Reason for the Velocity Decrease After Soaking

Figure 1 shows that after soaking cells in droplets, their average velocity in nanoliter volumes was reduced by about 40% compared to those soaked in large volumes (10 nL and more). Two hypotheses are proposed: (i) physiological deterioration due to an increase in salt concentration caused by water diffusion from the droplet, and (ii) deceleration due to the increase in viscosity.

The following physical model is used to describe the expected velocity decrease resulting from the increased viscosity. There are two forces acting on a sperm cell: the driving force of the flagellum [23] in the direction of movement and the drag force of the viscosity in the opposite direction. The sperm cell can be approximated as a tri-axial ellipsoid having axes *a* = 2.5 µm, *b* = 1.0 µm, and *c* = 1.5 µm.

Based on previous work [24], it can be assumed that sperm cells move at a constant average velocity, so these two forces are of equal magnitude. Stokes’ Law for an ellipsoid is [25](18)$$\begin{array}{c} F=6\pi \eta v\sqrt{b\cdot c}K \end{array}$$
where *F* is the drag force, v is the involuntary velocity of the sperm cells, *η* is the viscosity, *b*, *c* are the semi-axes perpendicular to the direction of movement, and *K* is the shape factor of an ellipsoid [25], which is equal to(19)$$\begin{array}{c} K=\frac{\frac{4}{3}\left(\beta^{2}-1\right)}{\frac{\left(2\beta^{2}-1\right)}{\sqrt{\left(\beta^{2}-1\right)}}\ln\left(\beta +\sqrt{\beta^{2}-1}\right)-\beta } \end{array}$$
where *β* is the ratio between the major axis and the geometrical mean of the other two minor axes (i.e., $\beta =a/\sqrt{b\cdot c}\approx 2.08$), yielding K≈1.22. Using the measured viscosities of the sperm cell suspension ($\eta_{s,0}$, see above, Section 2.4) and velocities ($v_{0}$, Figure 1), Equation (18) can be used to determine the drag force on the sperm cell (F). Since the driving force of the flagellum is approximately constant in the absence of physiological phenomena, at any given time, the drag force *F* remains constant and equal to the driving force. The viscosity of the droplet after 30 min was determined (see above), and this provides an estimate of the cell velocity under the assumed absence of physiological phenomena:(20)$$\begin{array}{c} v=\frac{F}{6\pi \eta \sqrt{b\cdot c}K} \end{array}$$

Figure 4 shows the measured velocities compared to the estimates from Equation (20). There is good agreement between the two for larger initial volumes, but at smaller volumes, there is a very significant and increasing deviation of the model from the measured velocities. This indicates that viscosity alone cannot explain the deceleration and clearly physiological phenomena—such as increased salt concentration—negatively affecting the sperm cells.

### 2.6. Preventing Sperm Cell Dehydration in Nanoliter Volumes

Consider Fick’s first law in Equation (10). The diffusion coefficient (*D*) depends solely on the behavior of the molecules at the oil–water interface [26], which is contingent on the host properties and apparently unchangeable in any given system. The rate of diffusion (*j*) is proportional to the concentration gradient across the droplet surface $\left(\frac{dC}{dS}\right)$. In principle, reducing this gradient should reduce the rate of diffusion and, as a result, increase the survivability of the sperm cells within the droplet. To test this, an oil–water mixture was stirred for 1 h, and the saturated oil layer was separated and used for cryopreservation. In a previous study, we showed that the solubility of water in oil is very low but not insignificant [27].

## 3. Discussion

The cryopreservation of small numbers of sperm cells—or even one—presents special challenges. These samples are at a nanoliter scale, but typical preparation methods use a protective oil layer over the nanoliter droplet to protect it from evaporation, which increases with decreasing droplet radius. Unfortunately, we observed that, in these nanodroplets, the motility and velocity of the sperm cells are dramatically decreased (Figure 1). In volumes below 30 nL, sperm cells do not survive the freeze–thaw cycle. Two effects were proposed to explain this phenomenon: increased viscosity and increased salt concentration resulting from water diffusion from the droplet. Our modeling of the effect of viscosity on sperm cell velocity showed that this alone cannot explain the deceleration of the sperm cells. This means that the diffusion of water from the droplet causes a significant increase in salt concentration, which has a deleterious effect on the sperm cells.

In a previous study [27], we showed that contrary to common perception, water has low solubility in oil. Likewise, the rate of diffusion of water from the droplet is proportional to the concentration gradient across the droplet surface. In larger droplets, a small amount of diffusion will have a small effect on salt concentration, but in a nanodroplet, this same amount of diffusion will have a more dramatic impact because the proportion of water lost to the amount remaining is larger.

Thus, we propose a simple, user-friendly approach to safely cryopreserve small droplets of sperm cells. By substituting the protective covering layer of oil with water-saturated oil, we were able to achieve a remarkable improvement in cell motility and velocity. While the focus of this study was the cryopreservation of sperm cells, it is reasonable to assume that the conclusion here can be applied to the cryopreservation of other biological samples.

The novelty and significance of this manuscript include the identification of water diffusion as the primary cause of cell death in nanoliter cryopreservation and the development of a simple, effective method using water-saturated oil to overcome this challenge, demonstrating significantly improved cell survival.

It is important to contextualize the scope of the present study. The simplified droplet-in-oil model was deliberately chosen to isolate and understand a fundamental biophysical challenge—catastrophic cell death at the nanoliter scale—which we first observed during the development of a novel cryopreservation device. The primary goal here was therefore not to develop a new automated workflow, but to diagnose the problem and validate a solution. Having now identified water diffusion as the mechanism, the principle of using water-saturated oil can now be integrated into more advanced cryopreservation platforms to establish robust and effective Standard Operating Procedures (SOPs).

While our approach shows significant promise, we recognize its limitations. First, the precise level of water saturation in the oil may vary slightly between preparations, potentially introducing minor variability, although our results indicate the method is robust. Second, our study focused on the biophysical mechanism of water diffusion and immediate post-thaw survival, assessed by motility. The long-term viability and fertilizing potential of sperm cryopreserved with this method require further investigation through embryonic development studies. Finally, while we propose this method could be applied to other cell types, its efficacy for cells with different sizes and membrane permeability characteristics would need to be empirically validated.

The broader impact is the potential to revolutionize fertility treatments for patients with limited sperm samples and applicability to the cryopreservation of other biological materials, advancing practices in reproductive medicine and biosciences.

This investigation was intentionally designed to be a foundational study, prompted by a puzzling phenomenon we observed while developing a novel single-sperm cryopreservation device, which was recently published [28]. In that work, we encountered zero sperm survival in nanoliter volumes, a stark contrast to microliter volumes using the same device.

To isolate and understand the destructive physical mechanism at play, separate from the complexities of the new device, we deliberately employed the simplified model system presented here—droplets in oil. Therefore, the goal of this study was not to develop a new automated tool, but rather to first deeply diagnose the cause of cell death.

## 4. Materials and Methods

### 4.1. Materials

Spermatozoa (about 20 samples from anonymous donors) were obtained from the Infertility and IVF Unit of the Chaim Sheba Medical Center, Ramat Gan, Israel. The recruitment period for this study was 1 August 2019 to 31 December 2022. Quinn’s Advantage Sperm Freezing Medium (CooperSurgical, Ballerup, Denmark) and oil for tissue culture were purchased from Sage (SAGE In Vitro Fertilization, Målov, Denmark).

### 4.2. Ethics

Sperm was obtained after receiving signed informed consent from each patient to use cells that would have otherwise been discarded. All personal data were fully anonymized to maintain patient privacy, and samples were coded so that patient information could not be retrieved. This study was approved by the local Institutional Review Board (approval number 0187-23-SMC).

### 4.3. Measurement System

Images were acquired using a motorized Olympus inverted IX81 microscope (Tokyo, Japan), which is equipped with a sub-micron Marzhauser Wetzlar motorized stage type SCAN-IM, with an L step controller (Wetzlar-Steindorf, Germany) and a filter wheel including a fluorescein fluorescence cube containing excitation filters (470–490 nm), dichroic mirrors (505 nm long pass), and emission filters (510–530 nm). The filters were obtained from Chroma Technology Corp. (Brattleboro, VT, USA). A cooled, highly sensitive 14-bit, ORCA II C4742-98 camera (Hamamatsu, Japan) was used for imaging. Olympus Cell^P software version 1.7 (Tokyo, Japan) was used for image analysis.

### 4.4. Freezing and Thawing Sperm Cells

Spermatozoa, washing medium, and freezing medium were mixed in a 1:1:2 ratio. Using a pipette, 0.1 µL droplets were deposited under the oil in a glass-bottom Petri dish. With the pipette tip under the microscope, the droplet was broken up into smaller, random droplets. Droplets with circular cross-sections were selected, and their exact volume was calculated (see below).

Droplets of varying volumes of freezing medium containing sperm cells were injected under oil onto a Petri dish. Using software developed in-house. The survival rates and average velocity of the sperm cells in the droplets were measured at three time points: (a) immediately after the formation of the droplets (initial state), (b) 30 min later, and (c) after the freeze–thaw cycle. The initial state was measured for review and comparison. Initial motility was used to normalize the value of percent motility in subsequent measurements.

The Petri dish was placed in liquid nitrogen vapor for 5–10 min and then immersed in liquid nitrogen. In the thawing process, the Petri dish was transferred to a microscope and incubated at 37 °C.

A total of 20 semen samples were obtained from individual anonymous donors. For the statistical analysis of sperm survival post-thaw, we compared the outcomes in droplets smaller than 30 nL with those in droplets larger than 30 nL. Due to the binary nature of the outcome (i.e., cells either survived or did not) and the presence of a zero-count in the smaller droplet group, Fisher’s Exact Test was employed. This test is ideal for comparing proportions between two groups and is highly accurate for small sample sizes or when counts are zero, as was the case in our experiments. The analysis confirmed that the difference in survival rates was extremely statistically significant, with a *p*-value of <0.0001. The clarity of this result demonstrates a definitive threshold effect. Because the observed phenomenon was so pronounced, this direct and robust test was deemed sufficient to confirm the finding.

### 4.5. Cell Count and Viability/Motility Assessment

To obtain the best statistical information for dozens or hundreds of motile sperm cells within a nanoliter droplet, we developed unique software based on machine learning with neural networks (the software is available from the authors upon request). The software takes a video clip of the sample under the microscope, learns to identify sperm cells, tracks them with an accuracy close to 100%, and reports the percentage of motile cells and their average velocity at the single-cell level. The software tracks sperm movement paths across sequential image frames. It removes immobile or abnormally mobility sperm, enhancing data quality and focus on viable cells. Involuntary movements deviating significantly from the sperm’s axis are identified and quantified.

### 4.6. The Viscosity Measurement of Macroscopic Volume Media

The viscosity of macroscopic volumes was measured with a HAAKE MARS 60 Rheometer (ThermoScientific, Dreieich, Germany). In this device, the sample liquid is placed within the annulus formed by one cylinder located inside a second cylinder. One of the cylinders is rotated at a set speed, which determines the shear rate inside the annulus. The liquid tends to drag the other cylinder, and the force (torque) it exerts on the second cylinder is measured and converted to shear stress. Two speeds give two points on the ‘flow curve’ [29], which is sufficient to define a Bingham plastic model [30] that is widely used in the oil industry for determining the flow characteristics of drilling fluids [31].

### 4.7. Assessment of the Viscosity of a Microscopic Fluorescent Droplet

Fluorescent (Fluorescein, Sigma, St. Louis, MO, USA) microscopic solution droplets were excited with polarized light (470 nm). The parallel $\left(I_{∥}\right)$ and perpendicular $\left(I_{⊥}\right)$ intensities of the polarized fluorescence emitted were measured using analyzer polarizers oriented parallel and perpendicular to the excitation vector field. The fluorescence polarization (p) is the ratio [16,32](21)$$\begin{array}{c} p=\frac{I_{∥}-I_{⊥}}{I_{∥}+I_{⊥}} \end{array}$$

The relation between the measured polarization $\left(p\right)$ and the intrinsic polarization $\left(p_{0}\right)$, the fluorescence polarization of frozen-like gas fluorescent solution, i.e., the fluorescent solution in which there is no rotational movement of molecules, is given by Perrin’s equation [33]: 1/p − 1/3 = (1/$p_{0}$ − 1/3) (1 + τ/$\tau_{r}$), where $\tau_{r}$ is the rotational correlation time. For quasispherical fluorophores, it is $\tau_{r}=\frac{\eta V}{RT}$ [34,35,36]. In this manner, the viscosity from fluorescence measurements can be determined, where the viscosity is(22)$$\begin{array}{c} \eta =\frac{\tau RT\rho }{M_{W}}\frac{\left(\frac{1}{p_{0}}-\frac{1}{3}\right)}{\left(\frac{1}{p}-\frac{1}{p_{0}}\right)} \end{array}$$

### 4.8. The Volume of a Microdroplet

The relation between the volume of a droplet on a surface and its cross-sectional area A, as derived in the Appendix A, is(23)$$\begin{array}{c} V\left(A\right)\approx 0.145\cdot A^{3/2} \end{array}$$

The contact angle (α) of 51° at room temperature, which was used in determining the coefficient in Equation (3), was measured using an OCA20 goniometer (DataPhysics Instruments GmbH, Filderstadt, Germany). The cross-sectional area was measured using image analysis software included with the microscope. The accuracy of this model was tested by injecting known volumes of water onto a glass surface. The obtained relation of $V\left(A\right)=0.143\cdot A^{1.57}$ is sufficiently close to Equation (3), and the slight deviation could be explained by the level of accuracy in measuring α [37,38].

### 4.9. Preparing the Saturated Oil

The water-saturated oil was prepared based on the protocol established in our prior research [28]. Specifically, sterile, tissue-culture grade mineral oil (Sage In Vitro Fertilization, Målov, Denmark) and nuclease-free water were mixed at a 10:1 oil-to-water volume ratio in a sterile container. The mixture was stirred continuously at 600 rpm for 1 h at room temperature (25 °C) to facilitate maximal water dissolution into the oil phase. Following stirring, the mixture was left to stand for 30 min to allow for complete phase separation. The upper, water-saturated oil layer was then carefully pipetted for use in the cryopreservation experiments. Saturation was confirmed when a distinct water phase remained at the bottom of the container, indicating the oil could not dissolve additional water.

## Figures and Tables

**Figure 1 ijms-26-06343-f001:**
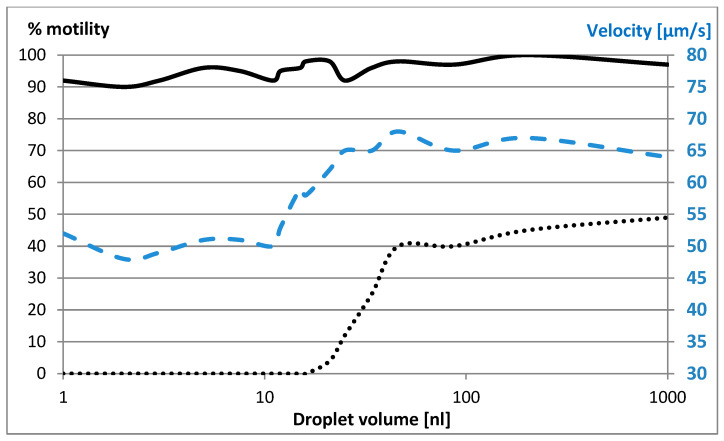
% motility is the survival rate, which is measured by motility at a given time divided by initial motility (left axis). The solid line represents motility after 30 min. The dotted line represents motility after the freeze–thaw cycle. Velocity (right axis) represented by the dashed line indicates the average velocity after 30 min.

**Figure 2 ijms-26-06343-f002:**
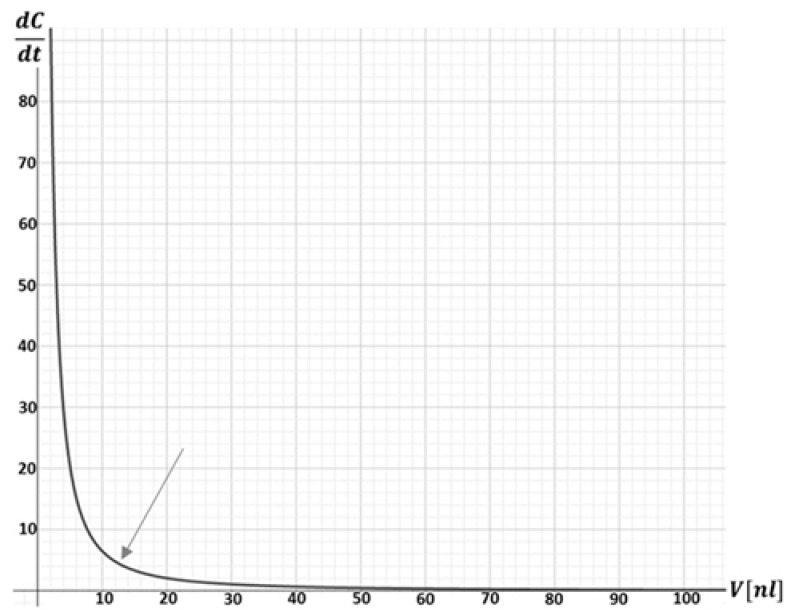
The rate of solute concentration changes (mol/(nL⋅min)) vs. droplet volume (nL). A sudden leap is clearly observed in small nanoliter volumes (gray arrow) graph prepared using https://www.symbolab.com/graphing-calculator software (accessed on 28 June 2022).

**Figure 3 ijms-26-06343-f003:**
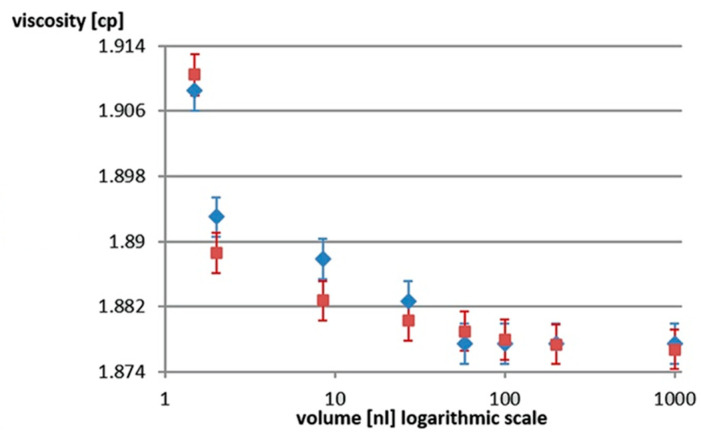
Calculated viscosity after 30 min by two different methods: the increase in fluorescence polarization (blue diamonds) and the decrease in volume measured by bright field (red squares).

**Figure 4 ijms-26-06343-f004:**
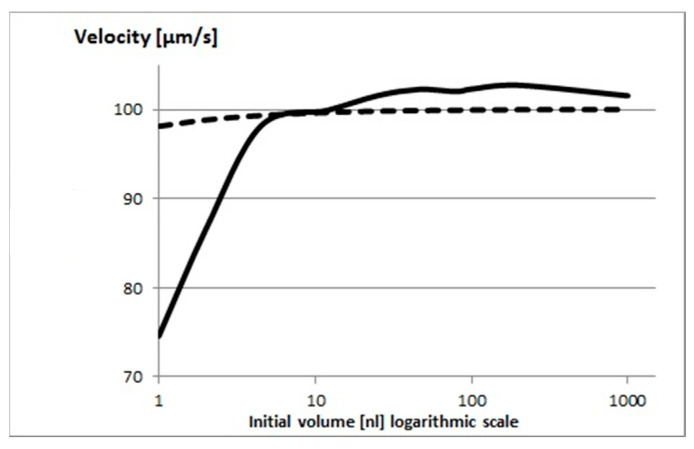
The estimated (dashed line) and measured (continuous line) velocities (µm/s) of sperm cells in droplets after 30 min (at room temperature) versus initial droplet volume. The velocities were normalized by the initial velocities $\left(v/v_{0}\right)$.

## Data Availability

All data generated or analyzed during this study are included in the manuscript and the Appendix A.

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
