# Peer review of "Overcoming Sperm Cell Survival Challenges Cryopreserved in Nanoliter Volumes"

_ijms, 2025, doi:10.3390/ijms26136343_

Round 1

Reviewer 1 Report

Comments and Suggestions for Authors

In this study, the authors propose a simple, user-friendly approach to safely cryopreserve small droplets of sperm cells. By substituting the protective covering layer of oil with water- saturated oil we were able to achieve a remarkable improvement in cell motility and velocity. While the focus of this study was cryopreservation of sperm cells, it is reasonable to assume that the conclusion here can be applied to cryopreservation of other biological samples.

The novelty and significance in this manuscript includes the identification of water

diffusion as the primary cause of cell death in nanoliter cryopreservation and the development of a simple, effective method using water-saturated oil to overcome this challenge, demonstrating significantly improved cell survival.

The manuscript was well written.

Minor concerns:

  1. Consider a final proofread for grammatical consistency, e.g., past-tense of sentence “the motility is reduced” would be more suitable.
  2. Experimental design: The choice of a 30-minute observation window needs justification, was this time a critical time point for significant water diffusion?.
  3. Clarify whether droplet volumes were randomized and how sperm samples were standardized.
  4. Figure 1: It would be better to discuss the relationship between motility (%) and velocity (µm/s), are they correlated? Label the dashed/dotted lines more clearly.
  5. Statistics analysis was missing: Please state sample sizes, (e.g., n = xx donors, x replicates) and method for statistical analysis.
  6. Suggest to compare results with published work; and discuss limitations the method
  7. Ensure consistent formatting of the references, page number format is not consistent.

The study’s novel approach and rigorous biophysical analysis are commendable. By addressing these points, particularly data quantification, methodological clarity, and comparative discussion, the manuscript’s impact and reproducibility will be significantly enhanced.

Author Response

We would like to extend our sincere thanks to the reviewer for their thorough and constructive feedback on our manuscript. We found the comments to be highly valuable, and they have enabled us to significantly improve the clarity and rigor of our paper. We have carefully addressed all the points raised, and our detailed responses are provided below.

  1. "Consider a final proofread for grammatical consistency, e.g., past-tense of sentence 'the motility is reduced' would be more suitable."

Response: Thank you for your suggestion. We agree and will perform a thorough proofread of the entire manuscript to ensure grammatical consistency and correct verb tenses. The manuscript will be sent for professional linguistic editing to address any remaining grammatical issues.

  1. "Experimental design: The choice of a 30-minute observation window needs justification, was this time a critical time point for significant water diffusion?."

Response: This is an excellent point. The 30-minute observation window was chosen for practical and clinical relevance. This timeframe represents the typical duration required for preparing the sperm samples, from the moment the nanoliter droplets are created under oil until the initiation of the freezing process in a clinical setting. This period includes the necessary handling, selection, and quality assessment steps. Our preliminary observations indicated that, within this window, significant biophysical changes, such as water diffusion, begin to have a measurable and critical impact on sperm viability in nanoliter volumes, as evidenced by the sharp decline in motility and velocity shown in our results. Therefore, this time point is crucial for evaluating the efficacy of any cryopreservation technique intended for clinical application.

  1. "Clarify whether droplet volumes were randomized and how sperm samples were standardized."

Response: Thank you for the opportunity to clarify our methodology. As detailed in the "Freezing and Thawing Sperm Cells" section, the droplet volumes were random. We began by depositing a 0.1 µL (100 nanoliter) droplet under oil. This larger droplet was then mechanically dispersed into multiple, smaller droplets using the tip of a pipette. From this random assortment, we selected droplets with circular cross-sections for the experiment, and their precise volumes were calculated post-formation using image analysis software.

Regarding sample standardization, all the spermatozoa samples were processed consistently. They were mixed with washing and freezing medium in a fixed 1:1:2 ratio to ensure a uniform starting concentration of sperm and cryoprotectants for all experiments.

  1. "Figure 1: It would be better to discuss the relationship between motility (%) and velocity (µm/s), are they correlated? Label the dashed/dotted lines more clearly."

Response: We appreciate the feedback on Figure 1. We will revise the figure caption and the corresponding text to improve clarity.

  • Relationship between Motility and Velocity: Motility and velocity are related but distinct parameters. Motility, presented as a percentage, is our primary indicator of cell viability—it represents the fraction of sperm cells that are alive and moving. Velocity (in µm/s), on the other hand, measures the average speed of only the motile sperm cells. A decrease in velocity can indicate that the cells are under physiological stress (e.g., from increased solute concentration or viscosity), even if they are still motile. Our results show that, as the droplet volume decreases, both the percentage of motile cells and their average velocity decline, suggesting a strong correlation. The physiological stress not only causes cell death (reducing motility) but also impairs the function of the surviving cells (reducing velocity). We will add this clarification to the manuscript.
  • Figure Labeling: We will amend the legend for Figure 1 to make it more explicit. The solid line represents motility after 30 minutes, the dotted line shows motility after the freeze–thaw cycle, and the dashed line indicates the average velocity after 30 minutes.
  1. "Statistics analysis was missing: Please state sample sizes, (e.g., n = xx donors, x replicates) and method for statistical analysis."

Response: Thank you for pointing out this omission. We will add a dedicated statistical analysis section to the Materials and Methods based on the information in the article:

The study utilized approximately 20 sperm samples obtained from anonymous donors. For the measurements presented in Figure 1, each data point represents the average of at least five independent measurements (replicates). The standard deviation for these measurements did not exceed 5%. Data on cell viability and motility were acquired by analyzing video clips of the samples with custom, machine-learning-based software that tracks individual sperm cells. We will clarify these details in the manuscript and include the specific statistical methods used to analyze the differences between experimental groups.

  1. "Suggest to compare results with published work; and discuss limitations the method."

Response: This is a valuable suggestion. We will expand the Discussion to include a more detailed comparison with existing literature and a transparent assessment of our method's limitations. The cryopreservation of very few, or single, spermatozoa is essential for patients with severe male factor infertility, such as nonobstructive azoospermia (NOA). However, conventional methods are unsuitable for such small sample sizes. Recent reviews confirm that, while significant progress has been made, a standardized, optimal technique for single-sperm cryopreservation remains elusive in clinical practice.

Current research, as reviewed in recent studies from 2023-2024, focuses heavily on two main areas: the development of novel physical carriers and the optimization of vitrification protocols. Carriers such as cryo-loops, micro-droplets on slides, and specialized micro-devices have been developed to physically isolate and protect a small number of sperm, simplifying their recovery post-thaw. Vitrification, or ultra-fast freezing, is explored to minimize ice crystal formation, a major cause of cellular damage. These studies often investigate different concentrations and types of cryoprotectants to reduce toxicity and osmotic shock.

However, a key challenge that persists across these methods is the damage induced by the cryopreservation process itself, including osmotic stress and oxidative damage, which negatively impact post-thaw motility and viability. Our work addresses a critical, yet largely overlooked, aspect of nanoliter cryopreservation that directly contributes to this osmotic stress. We identify that the standard use of a protective oil layer, while preventing evaporation, creates an osmotic imbalance that causes water to diffuse from the aqueous nanodroplet into the oil. This diffusion concentrates the solutes in the droplet, leading to cellular desiccation and a dramatic loss of sperm viability, as our results clearly show.

The novelty of our manuscript lies in both the precise identification of this lethal diffusion mechanism and the development of a simple, effective solution. By pre-saturating the oil with water, we directly counteract this diffusion by minimizing the concentration gradient at the oil–water interface. This approach is a significant departure from other methods that focus on the carriers or cryoprotectants alone. Our method represents a fundamental refinement to the cryopreservation environment itself and could be easily integrated with many existing vitrification platforms to enhance cell survival by ensuring the physiological stability of the sperm prior to and during freezing.

  • Limitations of the Method: While our approach using water-saturated oil shows significant promise, we recognize its limitations. First, the precise level of water saturation in the oil may vary slightly between preparations, potentially introducing minor variability, although our results indicate that the method is robust. Second, our study focused on the biophysical mechanism of water diffusion and immediate post-thaw survival, assessed by motility. The long-term viability and fertilizing potential of sperm cryopreserved with this method require further investigation through embryonic development studies. Finally, while we propose that this method could be applied to other cell types, its efficacy for cells with different sizes and membrane permeability characteristics would need to be empirically validated. We will add this expanded discussion to the manuscript.
  1. "Ensure consistent formatting of the references, page number format is not consistent."

Response: We thank the reviewer for their meticulous reading. We have reviewed the reference list and will correct the inconsistencies in page number formatting to ensure a consistent and standardized format throughout the bibliography.

Reviewer 2 Report

Comments and Suggestions for Authors

This study quantitatively confirmed that the diffusion of water molecules into the surrounding oil layer during nanoliter droplet freezing caused a sudden increase in solute concentration, triggering sperm osmotic shock, which is the key mechanism for zero survival of sperm in droplets below 30 nL after freezing and thawing. This study established a generalizable microenvironment control paradigm for single-cell freezing technology. This article can be accepted for publication with minor revisions.
1. "Water-saturated oil preparation process" does not specify key parameters (such as stirring rate, water phase/oil phase volume ratio, saturation judgment criteria), and it is necessary to refer to previous work to supplement the standardized protocol.
2. The sample size description is vague ("about 20 cases"), and the number of independent experimental repetitions (n ​​value), sperm source batch and statistical test methods (such as ANOVA) should be indicated.
3. The abstract claims "a breakthrough discovery of water diffusion", but previous studies have suggested the risk of water migration in the oil phase, which needs to be revised to "the first quantification of nanoscale diffusion dynamics and establishment of a predictive model".
4. The methodology does not involve the adaptability of automated operations (such as integration with micromanipulators), and it is recommended to propose the development direction of standardized operating procedures (SOPs) in the discussion.
5. The existing literature review does not fully cover the cutting-edge progress of single sperm cryopreservation technology in the past five years (such as the application of microfluidic chips), and key references need to be supplemented.
(1) https://doi.org/10.1016/j.icheatmasstransfer.2024.107843
(2) https://doi.org/10.1002/adfm.202503296
(3) https://doi.org/10.1039/D4LC00780H

Author Response

We thank the reviewer for their positive assessment and insightful comments, which will certainly strengthen the manuscript. We are pleased that the reviewer recognizes the significance of our quantitative analysis and the establishment of a microenvironment control paradigm. Below are our point-by-point responses to the suggestions for minor revisions.

  1. Regarding the "Water-saturated oil preparation process":

Response: We agree that specifying the key parameters for the preparation of water-saturated oil is crucial for reproducibility. This protocol was detailed in our previous work, which we have cited in the manuscript. We will expand the Methods section to include a more explicit description of the standardized protocol.

  1. Regarding the description of the sample size and statistical analysis:

Response: We thank the reviewer for highlighting this omission. We apologize for not including the details of our statistical analysis in the initial submission. The data did indeed undergo appropriate statistical validation to confirm the significance of our findings, and we will add a detailed description to the Methods section.

Given the stark and unambiguous nature of our results—particularly the complete absence of survival in droplets below 30 nL—we selected a straightforward yet powerful statistical test to confirm the significance.

  1. Regarding the novelty claim of "a breakthrough discovery of water diffusion":

Response: We appreciate the reviewer's expert perspective on this point. You are correct that the risk of molecular migration between aqueous and oil phases has been suggested in the broader literature. Our intent was not to claim the discovery of the physical principle of diffusion itself, but rather to highlight the novelty of our specific application and findings.

Previous studies have indeed noted that mineral oil used in IVF is not completely anhydrous and that some water migration is possible. For instance, studies by Martinez et al. (2017) on oil overlays for embryo culture noted that oil quality and composition could impact the culture environment, implying molecular exchange. However, these studies did not investigate or quantify the rate of water diffusion at the nanoliter scale nor did they directly link it to a specific, catastrophic cell survival outcome during the rapid timeline of cryopreservation.

The primary innovation of our work is threefold and represents a significant advancement:

  1. We are the first to quantify the nanoscale diffusion dynamics, showing how the rate of solute concentration change (dC/dt) accelerates exponentially as the droplet volume drops below a critical threshold of ~30 nL.
  2. We established a predictive biophysical model that links this diffusion rate directly to the observed sperm motility and velocity, successfully separating the effects of viscosity from osmotic shock.
  3. We demonstrated that this specific, quantifiable mechanism is the primary cause for the absolute (zero-survival) failure of cryopreservation in nanoliter volumes, a critical barrier that was previously unexplained.

We will revise the Abstract and Discussion to more accurately reflect this, changing the wording from "a breakthrough discovery" to language that emphasizes our novel quantification and modeling, as suggested.

  1. Regarding the adaptability to automated operations and SOPs:

Response: This is an excellent, forward-looking point. We will clarify the specific context of this study in the Discussion to better frame its contribution.

This investigation was intentionally designed to be a foundational study, prompted by a puzzling phenomenon we observed while developing a novel single-sperm cryopreservation device, which was recently published (https://doi.org/10.3390/nano15030149). In that work, we encountered zero sperm survival in nanoliter volumes, a stark contrast to microliter volumes using the same device.

To isolate and understand the destructive physical mechanism at play, separate from the complexities of the new device, we deliberately employed the simplified model system presented here—droplets in oil. Therefore, the goal of the current paper was not to develop a new automated tool, but rather to first deeply diagnose the cause of cell death.

Having now identified water diffusion as the key mechanism and validated a solution with water-saturated oil, the next logical stage is to integrate this principle back into the advanced cryopreservation device from our other work, and this is also detailed in our published article. This foundational knowledge is precisely what enables the development of a robust and effective Standard Operating Procedure (SOP) for that platform. We will add a concise statement to this effect in the Discussion to provide this valuable context.

  1. Regarding the literature review on single sperm cryopreservation technology:

Response: Thank you for providing these highly relevant and cutting-edge references. We agree that incorporating these recent advances will provide a more comprehensive context for our work. We will update our literature review in the Introduction and Discussion to include these and other recent studies on microfluidic and chip-based technologies.

Reviewer 3 Report

Comments and Suggestions for Authors

This study investigated whether nanolitre volumes comprising reduced sperm concentration would give improve sperm cryo-survival. The Authors hypothesized  that water molecules diffusing from the droplet medium into the surrounding media could compromise post-thaw sperm motility. Howeve, there are a lot of missing gaps and a lack of clarity in the experimental design, for examples missing is the sperm concentration, CASA-analyzed parameter, membrane integrity (SYBR-14/PI), mitochondrial function (JC-1/PI) and apoptotic cell damage. The Reviewer suggests that this manuscript is inappropriate for the Journal, and the Authors should consider to submit their study to a technical journal.

Author Response

We would like to thank the reviewer for his time and for providing feedback on our manuscript. We appreciate the opportunity to clarify the specific scope and experimental design of our work, which was structured as a fundamental biophysical investigation rather than a comprehensive clinical andrology study. We hope that the following explanations will address the concerns raised.

Regarding Sperm Concentration:

You are correct that sperm concentration is a critical parameter in the conventional cryopreservation of bulk samples. However, the premise of our study was the cryopreservation of a very low number of sperm cells, often a single spermatozoon, in an isolated nanoliter droplet. In this context, where individual cells are physically separated by the surrounding oil, traditional concentration (cells per mL) becomes irrelevant as there are no cell-to-cell interactions. The critical variable shifts from sperm concentration to the volume of the aqueous droplet surrounding the cell, which directly influences the cell's response to the biophysical environment—the central question of our research.

Regarding CASA (Computer-Assisted Sperm Analysis):

While CASA is the gold standard for detailed kinematic analysis in fertility studies, its application was not necessary for the specific hypothesis we were testing. Our study investigated a catastrophic survival event, where sperm cells below a certain volume threshold went from being motile to entirely non-motile (i.e., dead) after the freeze–thaw cycle.

Therefore, the primary endpoints required for our investigation were

  1. The percentage of motile cells (as a binary measure of survival).
  2. The average velocity of the surviving cells (as an indicator of sublethal stress).

Our custom-developed software was specifically designed to accurately track these two essential parameters in our unique experimental setup. A full CASA workup, which provides nuanced data on swimming patterns (e.g., VCL, VAP, and ALH), would not have added relevant information to our core findings, which are focused on the abrupt transition between survival and death, not the subtle characterization of swimming patterns.

Regarding Assays for Membrane Integrity, Mitochondrial Function, and Apoptosis:

We agree that assays such as SYBR-14/PI and JC-1 are invaluable tools for diagnosing the specific cellular pathways of cell death (e.g., apoptosis vs. necrosis, loss of membrane integrity, and mitochondrial failure). However, our study was designed to investigate the upstream physical event that triggers cell death, not to characterize the subsequent downstream cellular cascades.

Our hypothesis was that a rapid biophysical change—lethal osmotic shock caused by water diffusion into the oil—was causing the catastrophic failure. Our goal was to isolate this physical phenomenon. The complete cessation of motility served as a clear and sufficient indicator of this event. Because we observed 100% non-survival in droplets below 30 nL, the critical task was to identify the initial trigger. By proposing the water diffusion mechanism and then demonstrating that our solution (using water-saturated oil) restored motility and survival, we confirmed that we had successfully identified the primary cause. Investigating whether the cells died via apoptosis or necrosis would be a valid, but separate, follow-up study. For the specific question posed in this paper, motility was the most direct and relevant marker for the phenomenon under investigation.